# A Viscoelastic Model to Evidence Reduced Upper-Limb-Swing Capabilities during Gait for Parkinson's Disease-Affected Subjects

**Luca Pietrosanti** [1], **Cristiano Maria Verrelli** [1], **Franco Giannini** [1], **Antonio Suppa** [2,3], **Francesco Fattapposta** [2], **Alessandro Zampogna** [2], **Martina Patera** [2], **Viviana Rosati** [4] **and Giovanni Saggio** [1,*]

1   Department of Electronic Engineering, University of Rome Tor Vergata, 00133 Rome, Italy
2   Department of Human Neurosciences, Sapienza University of Rome, 00185 Rome, Italy
3   IRCCS Neuromed, 86077 Pozzilli, Italy
4   A.O.U. Policlinico Umberto I, 00161 Rome, Italy
*   Correspondence: saggio@uniroma2.it

**Abstract:** Parkinson's disease (PD) is a chronic neurodegenerative disorder with high worldwide prevalence that manifests with muscle rigidity, tremor, postural instability, and slowness of movement. These motor symptoms are mainly evaluated by clinicians via direct observations of patients and, as such, can potentially be influenced by personal biases and inter- and intra-rater differences. In order to provide more objective assessments, researchers have been developing technology-based systems aimed at objective measurements of motor symptoms, among which are the reduced and/or trembling swings of the lower limbs during gait tests, resulting in data that are potentially prone to more objective evaluations. Within this frame, although the swings of the upper limbs during walking are likewise important, no efforts have been made to reveal their support significance. To fill this lack, this work concerns a technology-based assessment of the forearm-swing capabilities of PD patients with respect to their healthy counterparts. This was obtained by adopting a viscoelastic model validated via measurements during gait tests tackled as an inverse dynamic problem aimed at determining the torque forces acting on the forearms. The obtained results evidence differences in the forearm movements during gait tests of healthy subjects and PD patients with different pathology levels, and, in particular, we evidenced how the worsening of the disease can cause the worsening of the mechanical support offered by the forearm's swing to the walking process.

**Keywords:** Parkinson's disease; motor impairment; TUG test; upper-limb swings

## 1. Introduction

Parkinson's disease (PD) is a chronic and progressive neurodegenerative disorder that affects the central nervous system. The disease manifests when there is a significant loss of dopaminergic cells in a region of the brainstem called the *substantia nigra.* The dopaminergic system plays a crucial role in motor control, and the decrement in dopamine levels in the brain is responsible for the genesis of its motor symptoms.

The cardinal motor symptoms of PD include tremor, bradykinesia (slowness of movement), rigidity (stiffness of muscles), dyskinesia (movements similar to tics or chorea) [1], and postural instability (impaired balance and coordination) [2,3]. These symptoms progressively worsen over time and can significantly impact a patient's quality of life.

In addition to motor symptoms, PD may also present non-motor symptoms [4], such as cognitive difficulties, speech impairments [5,6], mood changes, sleep disturbances, and autonomic dysfunction. These non-motor symptoms can vary widely among individuals and may appear before or after the onset of motor symptoms.

Among the most adopted protocols for PD assessment is the MDS-UPDRS scale [7], consisting of a 0–4-interval rating assessment of motor and non-motor symptoms, mostly

based on the qualitative rater's considerations [8] and therefore prone to intra- and inter-rater variability and experience [9], potentially mitigated by technology-based objective measurements. In such a frame, Lonini et al. [10] adopted inertial measurement units (IMUs) placed on the hand and classified the motor tremors and bradykinesia by means of a convolutional neural network (CNN) during the patient's daily activities. In a similar test, Di Biase et al. [11] explored the optimal configuration of IMUs for symptom detection in upper-arm tasks, demonstrating that, to discriminate between healthy and pathological subjects and between the ON and OFF medicine conditions, a single inertial unit placed on the distal location of the upper limb is sufficient. Cesarelli et al. [12] focused on motor tasks performed by the upper limbs to evidence kinematic features useful for classifying PD vs. healthy conditions through the so-called Knime Analytics Platform. Ricci et al. [13–15] employed a network of wearable IMUs and analyzed the collected data by means of the k-nearest neighbor (kNN) and Support Vector Machine (SVM) algorithms for assessing motor impairments in a PD de novo subject, and they found that the early stages of PD are characterized by a set of features that are not visible to the naked eye.

While researchers have been devoting efforts to objectively evaluate a number of different motor impairments, the particular (but crucial) aspect of the resistance to motion, which is rigidity [16], remains insufficiently analyzed. Rigidity in PD is characterized by limb stiffness, which is generally assessed by clinicians via the passive movements of the limbs.

This cardinal motor symptom greatly affects the daily lives of PD patients, contributing to their global motor slowness, gait impairment, and loss of independence. The objective evaluation of rigidity, better if performed remotely in an ecological setting, could improve clinical management with targeted interventions such as tailored therapies or focused physiotherapy.

We have recently shown that rigidity in PD can be objectively estimated by examining the frequency content of arm-swing movements during gait [17].

Arm-swing reduction is frequently observed in PD patients from the early stages of the disease, so it has been proposed as a prodromic marker of the disease. As an example, Mirelman et al. [18] adopted wearable sensors on the participant's trunk, ankles, and wrists during a gait test, and they found that 60% of the discriminative motor characteristics come from the wrists' motion only.

The pathophysiological mechanisms underlying reduced arm swing during gait in PD is still a matter of debate, but it has been demonstrated to be strongly associated with the severity of rigidity [17,19]. However, as far as we know, there are no validated methods to quantitatively assess rigidity in pathological subjects.

The present study aims to propose an easy-to-interpret model to analyze upper-limb mobility and impairments and quantitatively estimate rigidity in PD subjects from gait measurements.

To achieve this, here, we introduce an innovative descriptive model of the forearm swings during gait to underline the possible differences in the acting torque on the elbows of PD patients with respect to healthy subjects, and we quantify the reduced motor capabilities of the forearms by means of a finite set of coefficients representing biomechanical aspects of the forearm. In particular, the model refers to the Lagrangian formalism to describe the upper arms' swing and a double-pendulum system, taking a cue from works on healthy individuals [20–22], with elastic and viscosity properties embedded into the physical constraints.

This model is fed with data gathered from inertial sensors and is capable of computing information about the elastic and viscous properties of forearm muscles. Moreover, we aim to quantify the reduced arm swing in PD patients in order to examine possible clinical–behavioral correlations. We hypothesize that our model could be a new analytic tool for the instrumental assessment of patients with PD.

## 2. Materials and Methods

### 2.1. Subjects

A total of 37 volunteer participants were recruited at the Movement Disorders Outpatient Service of Sapienza University of Rome in Italy. Each participant was instructed on the procedure and gave informed consent according to the standard protocol of Helsinki. The participants were divided into two groups: one consisting of 18 PD patients and another consisting of 22 age-matched healthy subjects (HC, hereafter).

The clinical assessment included MDS-UPDRS scores, based on main motor symptoms' evaluation, including cardinal signs of PD (bradykinesia, rigidity, and tremor) and axial disorders (gait and balance issues), and H&Y scale, according to symptom distribution (symmetric/asymmetric involvement) or axial impairment. Cognitive functions were examined through MMSE, a multi-domain cognitive scale for evaluating several abilities such as orientation, memory, fluency, attention, visuospatial abilities, etc. These assessments provided valuable data for the analysis and comparison of the PD patients versus the control group, as well as for patient classification based on disease severity and clinical–behavioral correlations.

Inclusion criteria for PD patients were as follows: diagnosis of idiopathic PD based on current consensus criteria [23]; Hoehn and Yahr scale (H&Y) $\leq$ 2.5; ability to maintain an upright stance and walk independently. The exclusion criteria were as follows: diagnosis of atypical parkinsonism; severe PD (H&Y > 2.5); presence of L-Dopa-induced dyskinesia, dementia (as reflected by Mini-Mental State Examination—MMSE < 24) and comorbidities affecting gait or arm movements (including orthopedic or rheumatologic issues, polyneuropathies, and other neurological diseases).

PD patients were divided according to the severity of their upper limb rigidity as defined by item 3.3 of MDS-UPDRS-III. This item identifies five levels of severity: (0) no rigidity; (1) slight rigidity (only detected asking the patient to activate the contralateral limb); (2) mild rigidity (when full range of passive motion is easily achieved); (3) moderate rigidity (when full range of passive motion is achieved with effort); (4) severe rigidity (when full range of passive motion is not achieved at all). No patients with moderate or severe rigidity were enrolled, so the participants were divided into 3 groups (PD0, PD1, PD2) according to their 3.3 item of the MDS-UPDRS-III score (0, 1, 2, respectively).

PD patients were under chronic dopaminergic treatment. Arm swing during gait, as rigidity, bradykinesia, or tremor, can benefit from levodopa intake [19,24]. To observe rigidity without the influence of medication, we examined our patients when in the OFF-state of therapy, as intended after 12 h of L-Dopa withdrawal according to standardized procedures [25,26].

Concerning healthy subjects, only individuals not suffering from significant clinical disorders affecting gait and arm swing were enrolled.

Participants were matched for age and demographic characteristics (Table 1). Anthropometric data (mass, length, center of mass, and turning radius for both the upper arm and forearm) were computed from height and weight of each participant, as reported in [27], to feed the later detailed model (as in the following Section 2.4).

**Table 1.** Clinical and anthropometric features of patients with Parkinson's disease (PD) and healthy controls (HC).

|  | PD | HC |
|---|---|---|
| Age (years) | 65 $\pm$ 8 | 69 $\pm$ 11.2 |
| Gender | 12 M, 6 F | 10 M, 22 F |
| Height (cm) | 170.5 $\pm$ 11.5 | 163.1 $\pm$ 10.7 |
| Weight (Kg) | 73 $\pm$ 16.0 | 67.5 $\pm$ 12.4 |
| MDS-UPDRS III | 29.8 $\pm$ 10.7 |  |

MDS-UPDRS III: Movement disorder society—Unified Parkinson's disease rating scale part III.

### 2.2. Wearable Electronics

A number of different technologies have been developed and applied to gather movement data from subjects performing gait. These technologies are classified according to the locations of sources and sensors (if attached to the body being measured or elsewhere in the world, respectively [3]), or according to the type of adopted technology (basically: optical, electromagnetic, and physical ones). Within technologies, the camera-based systems are the most reliable and represent the gold-standard, but are expensive and require dedicated rooms and experienced technicians, whilst the wearable-based systems are less accurate but are low-cost, potentially ubiquitous, and offer an easy-to-use solution. In particular, the wearable based-systems refer to those electronics that can be unobtrusively embedded into clothes within supporting fabrics, such as bend of stretch sensors [28,29] or inertial measurement units (IMUs).

For our work, the motor capabilities of upper limbs were gathered by means of a network of five lightweight (<20 g for each) IMUs, termed Movit® (model G1, by Captiks Srl, Rome, Italy), previously successfully adopted [30,31] and validated against an optical-based gold-standard gait measuring system, and thus proven to perform more than sufficiently for our purposes [32]. Each IMU was placed on a specific district of the subject's body, that is, the upper back (thoracic vertebrae T7) and arms and forearms, by means of Velcro straps, as evidenced in Figure 1.

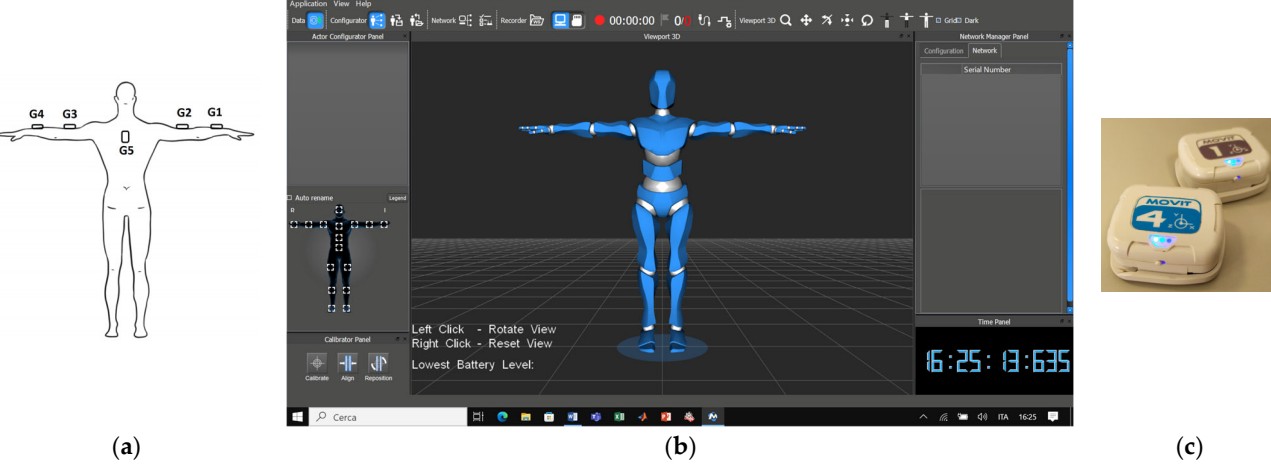

(**a**)  (**b**)  (**c**)

**Figure 1.** (**a**) The IMU sensors, termed Movit, labeled from G1 to G5, as seen on the back of the subject's body; (**b**) a screen of Motion Studio Software version 2.4 (capable to manage 16 IMUs at once) used for data acquisition; and (**c**) two Movit G1 units.

Each IMU (two examples are shown in Figure 1c) is equipped with a tri-axis accelerometer capable of measuring up to $\pm16$ g with 16.384 LSB/g and a tri-axis gyroscope with a measurement range up to $\pm2000°/$s and a resolution of 32.8 LSB/°/s [33,34]. The recording process is managed, at 200 Hz sampling frequency, by the Motion Studio software version 2.4 (a proprietary one, Figure 1b) running on a personal computer that receives data in wireless mode via ZigBee protocol [35]. The Motion Analyzer software version 1.2.98 (from the same manufacturer) furnishes the elbow angles (in degrees) from the gathered raw data.

### 2.3. Test Protocol

This study focuses on analyzing the forearm swings during gait performances gathered from the standard Timed Up and Go (TUG) test, a widely used protocol for evaluating gait impairments associated with various pathologies. The TUG test comprises multiple sub-phases (Figure 2a):

1. Sit to Walk: Starting from sitting, the participant gets up from the chair without the aid of the arms;
2. First linear walking: The subject walks (away from the chair) at a comfortable speed along a straight line for six meters;
3. Turning: The participant turns 180° around a pin to go back;
4. Second linear walking: Same as point 2, but in the opposite direction (toward the chair);
5. Stand To Sit: The participant turns themself around 180° and sits down.

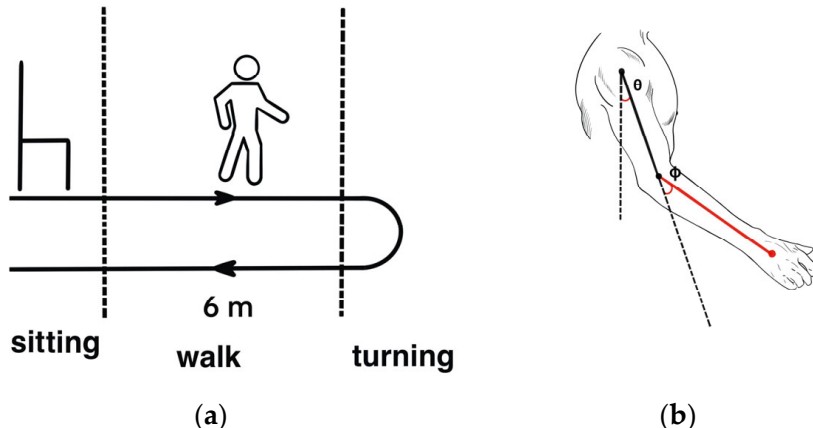

**(a)**  **(b)**

**Figure 2.** (**a**) Scheme of the Timed Up and Go (TUG) test and its sub-phases; (**b**) arm and forearm as a double pendulum system.

We intentionally asked each participant to follow the entire TUG protocol for completeness, being a standard test; however, our interest was focused on the linear walking phases only, and in particular on the upper arms' pendular movements in supporting the gait.

### 2.4. Double Pendulum Model

Arms' swings during gait are complex and not sufficiently explained tasks. Some studies [36,37] have shown their dependence on the activation and synergies of different muscles. By the way, for the purposes of the present study focused on the torque forces acting on the upper arms, we adopted a simple model based on the 2 link arm inverse kinematics. In particular, we modelled the arm as a double pendulum system [38] (Figure 2b). This model takes into account only the movement of the upper limb in the sagittal plane and does not consider wrist degree of freedom. Despite these simplifications, this model shows good agreement capabilities with the real data [39]. Accordingly, here, we express the dynamic behavior of the upper limb through the Lagrangian formulation, L being the Lagrangian function as the difference between kinetic and potential energies of the double pendulum, T and V, respectively:

$$L = T - V, \tag{1}$$

In particular, we model the two segments (arm and forearm) of the double pendulum adopting the anthropometric data previously computed (Section 2.1), with each segment characterized by mass $m_i$, length $l_i$, center of mass (with respect to the proximal joint) $r_i$, and moment of inertia $I_i$.

Considering the upper arm, the kinetic energy $T_1$, and the potential energy $V_1$ results in:

$$T_1 = \frac{1}{2} I_1 \theta^2, \tag{2}$$

$$V_1 = m_1 g r_1 (1 - \sin \theta), \tag{3}$$

while for the forearm, we obtain the following expression:

$$T_2 = \frac{1}{2}m_2 v_2^2 + \frac{1}{2}I_2(\dot{\theta} + \dot{\varphi})^2, \tag{4}$$

$$V_2 = m_2 g\left[l_1(1 - \sin\theta) + r_2(1 - \sin(\theta + \varphi))\right], \tag{5}$$

with $v_2$ being the center of mass velocity, defined by the following expression:

$$v_2^2 = (l_1\dot{\theta})^2 + 2l_1 r_2\dot{\theta}(\dot{\theta} + \dot{\varphi})\cos\varphi + (r_2\dot{\varphi})^2, \tag{6}$$

with $\theta$ and $\varphi$ being the angles formed by the arm and the forearm, respectively (Figure 2b), and $g$ the gravity acceleration.

The torque acting on shoulder and elbow joints can thus be obtained by differentiating the Lagrangian function as follows:

$$\frac{d}{dt}\frac{\partial L}{\partial\dot{\theta}} - \frac{dL}{d\theta} = \tau_\theta, \tag{7}$$

$$\frac{d}{dt}\frac{\partial L}{\partial\dot{\varphi}} - \frac{dL}{d\varphi} = \tau_\varphi, \tag{8}$$

where $\tau_\theta$ and $\tau_\varphi$ are the torques acting on the upper arm and the forearm, respectively. Specifically, and according to the MDS-UPDRS protocol, we focus on the elbow's dynamics and, consecutively, on forearm movements.

### 2.5. Dynamic of the Forearm

Mechanical properties of muscles can be represented as a system of springs and damping elements, as in Figure 3a, known as the Kelvin–Voight model [40,41]. According to this model, in this study, we express the torque acting on the elbow joint as a function of elastic and viscous elements, as depicted in Figure 3b:

$$\tau_\varphi = \beta_0 + \beta_1(\varphi_m - \varphi) + \beta_2\varphi + \beta_3\dot{\varphi}, \tag{9}$$

where the constant $\beta_0$ represents the forcing constant (due to muscle's activity); $\beta_1$ and $\beta_2$ represent the flexion and the extension elastic constants, respectively; $\beta_3$ represents the damping factor (due to the "viscosity" aspect of the muscle's movements); and $\varphi_m$ represents the maximum flexion angle of the elbow joint.

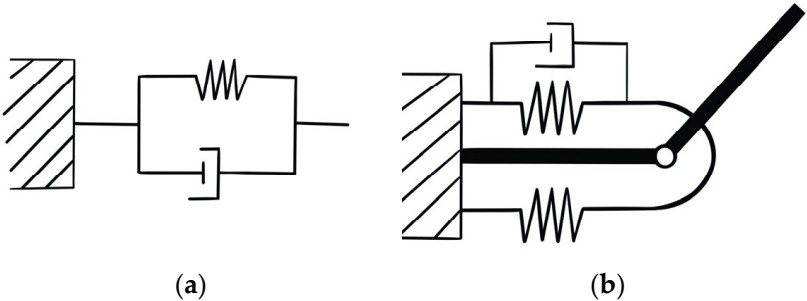

(**a**)          (**b**)

**Figure 3.** Representation of (**a**) single muscle with the Kelvin–Voight model and (**b**) biceps and triceps muscles as system of spring-damping elements.

### 2.6. Data Processing

In order to increase the signal-to-noise ratio of the analog signals gathered from the elbows' angles data during the TUG test, we approximated the measured data with an 8-term Fourier series, similarly to [1]. This step allows maintaining the significant information of the signal, removing unwanted frequencies and noise. Fourier series can be

applied to approximate periodic signals; however, if the signal loses its periodicity, Fourier series can occur in a loss of information. Due to the adopted measurement protocol and task, the signal recorded from arm oscillations are quasi-periodic, meaning that Fourier series can be applied with a negligible information loss. Smoothed data were inserted in the double pendulum model to obtain $\tau_\varphi$. The $\beta$ coefficients were then computed by fitting Equation (8) with measured $\tau_\varphi$ using the least-squares method. The optimization of $\beta$ coefficients was performed by means of a Matlab routine (by Mathworks Inc., Massachusetts, US) based on the trust-region-reflective algorithm. This procedure was first applied to healthy subjects to determine the mean reference values for $\beta$ features that characterize average values related to their swing capabilities. We then repeated the same procedure for each of the PD patients to evidence their differences with respect to the average values of their healthy counterparts, so as to underline the impact (if any) of the disease on the range of motion of the elbows' swings and on the elastic and damping aspects of the forearm movements for both upper limbs.

In particular, to highlight differences (if any) between the HC and PD groups of the $\beta$ coefficients, specifically related to elastic and dumping aspects of muscles acting on the forearm, we employed a Mann–Whitney U test (significant $p$-value set to 0.05).

Moreover, to evaluate any correlation between obtained data and clinical scores, we computed the Spearman Rank correlation coefficient (significant $p$-value set to 0.05) between the aforementioned $\beta_0$, $\beta_1$, $\beta_2$, $\beta_3$ coefficients and the score from item 3.3 of the MDS-UPDRS scale (particularly related to rigidity aspects of upper limbs). Spearman correlation was preferred instead of classical Pearson correlation because of the ordinal nature of UPDRS values.

## 3. Results

Figure 4a,b show the mean torque $\tau_\varphi$ values as obtained during a single complete oscillation of the forearm (starting when in maximum flexion) with the method previously detailed (Equation (8)), for both HC and PD groups, respectively.

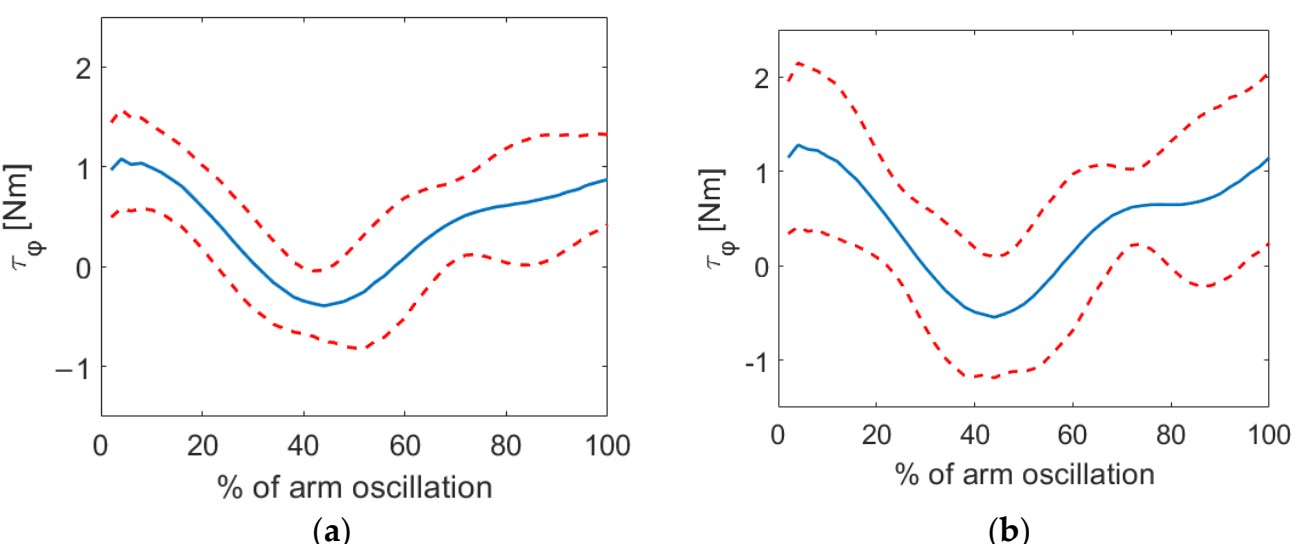

**Figure 4.** Average elbow torque (blue, solid lines) and related standard deviation (red, dashed lines) in a complete gait cycle for (**a**) HC group and (**b**) PD patients.

Figure 5 shows the RMSEs of the $\tau_\varphi$ values as determined by Equation (8) using the estimated four $\beta$ coefficients, with respect to the $\tau_\varphi$ value as obtained directly from the measurements, for the HC and PD group, respectively.

Statistics of the estimated coefficients are reported in Table 2 for both HC and PD groups, together with the results from the Mann–Whitney U test.

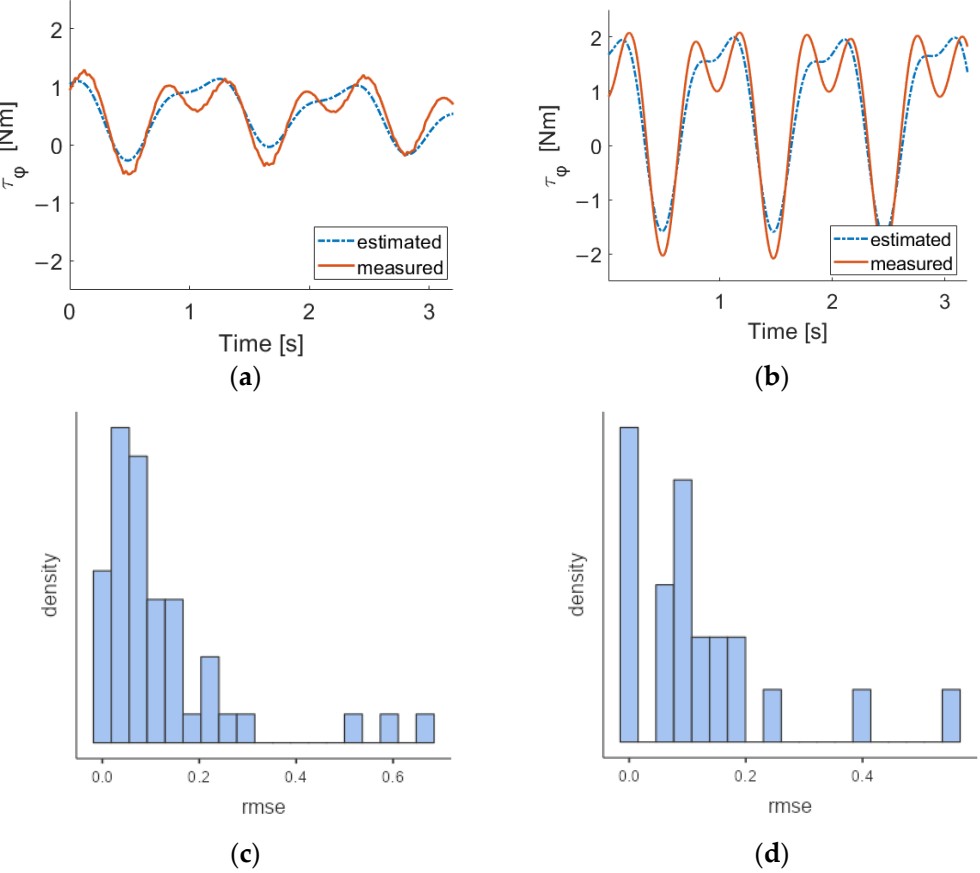

**Figure 5.** Torques measured (orange solid line) and estimated with $\beta$ coefficients (blue dashed line) for (**a**) HC and (**b**) PD subjects. Distribution of RMSE in (**c**) HC and (**d**) PD groups.

**Table 2.** Median and standard deviation (SD) values of beta coefficients for different PD subgroups, and results of the Mann–Whitney test. The meaningful *p*-values are evidenced in bold.

| | PD (MDS-UPDRS Item 3.3 Score) | | | HC (Median ± Sd) | *p*-Value | | |
|---|---|---|---|---|---|---|---|
| | PD (0) (Median ± Sd) | PD (1) (Median ± Sd) | PD (2) (Median ± Sd) | | | | |
| | | | | | PD 0 vs. HC | PD 1 vs. HC | PD 2 vs. HC |
| $\beta_0$ | $3.53 \pm 1.81$ | $2.56 \pm 1.71$ | $2.27 \pm 0.70$ | $2.69 \pm 1.02$ | 0.419 | 0.386 | **0.047** |
| $\beta_1$ | $-6.09 \pm 2.01$ | $-6.49 \pm 4.89$ | $-6.74 \pm 7.21$ | $-6.23 \pm 2.30$ | 0.760 | 0.792 | 0.348 |
| $\beta_2$ | $-15.0 \pm 5.31$ | $-12.3 \pm 6.85$ | $-13.6 \pm 9.63$ | $-11.6 \pm 3.88$ | 0.075 | 0.522 | 0.402 |
| $\beta_3$ | $-0.56 \pm 0.44$ | $-0.47 \pm 0.63$ | $-0.83 \pm 0.63$ | $-0.26 \pm 0.35$ | 0.166 | **0.049** | **0.040** |

Table 3 shows the results for Spearman's rank correlation between the $\beta_0$, $\beta_1$, $\beta_2$, $\beta_3$ coefficients of PD subjects and item 3.3 "limb rigidity" score.

**Table 3.** Spearman's rank correlations between $\beta$ coefficients and MDS-UPDRS item 3.3 score. The meaningful *p*-value is reported in bold.

| Feature | Rigidity (Sub-Item 3.3) | |
|---|---|---|
| | **r** | *p*-Value |
| $\beta_0$ | $-0.378$ | **0.041** |
| $\beta_1$ | $-0.155$ | 0.245 |
| $\beta_2$ | $0.116$ | 0.697 |
| $\beta_3$ | $-0.262$ | 0.119 |

Figure 6 reports the distribution of beta coefficients for different MDS-UPDRS item 3.3 scores.

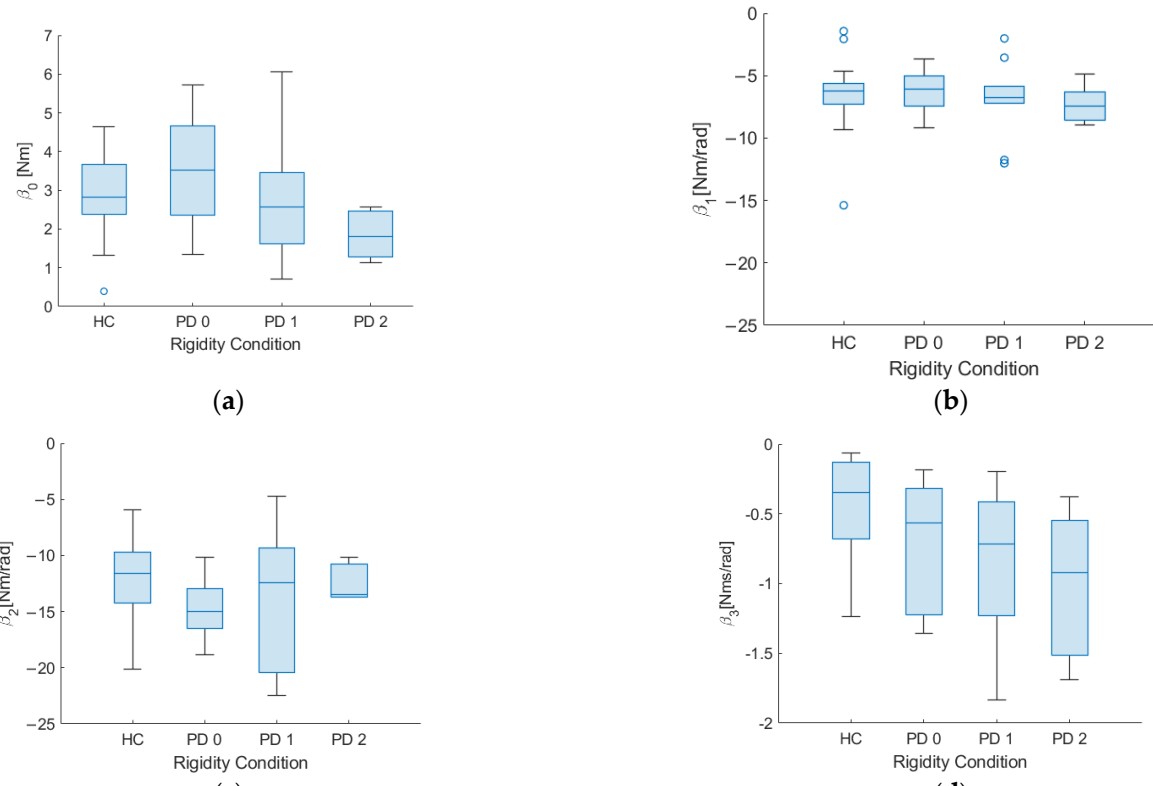

(a)

(b)

(c)

(d)

**Figure 6.** Distribution of (**a**) the forcing constant (because of the muscles' activities) $\beta_0$, (**b**) the flexion elastic constant $\beta_1$, (**c**) the extension elastic constant $\beta_2$, and (**d**) the dumping factor $\beta_3$ for different rigidity conditions. Circles outside the boxes represent outliers.

## 4. Discussion

In this work, we introduce and adopt a novel mathematical model to analyze the arm swing during linear walking to estimate the forces and rigidity acting on the elbow joints on the basis of measurements gathered during gait tests, with numerical coefficients representing the viscoelastic properties of the forearm.

Figure 4 illustrates the average curve depicting the relationship between the measured $\tau_\varphi$ and the percentage of arm oscillation. It is observed that when starting from a position of maximum flexion, the torque initially exhibits a positive value during the initiation of the forward oscillation phase. The point of maximum elbow flexion corresponds to roughly 50% of the oscillation, wherein the torque reaches its minimum value. Notably, in PD subjects, the $\tau_\varphi$ reaches lower values, opposing the forward movement, acting like a spring at its maximum extension trying to return to equilibrium position. In the second half of the arm oscillation, from maximum flexion to maximum extension, the torque returns to positive values. This can be attributed to the elbow blocking the backward oscillation of the forearm.

Figure 5a,b illustrate the torques extracted from the measurement values inserted into the double pendulum model (orange solid line), and the others as derived from β coefficients (blue dashed line) for, by way of example, but not limited to, one healthy subject (Figure 5a) and one PD patient (Figure 5b). In such a case, the extracted and derived coefficients are mostly coincident for negative values, which corresponds to the flexion phase, as explained for Figure 4, while for positive values, the model has a poor fit with the measured torque. This discrepancy can be addressed to the fact that we did not insert into our linear model (Equation (9)) a term to represent elbow constraint, which is characterized

by strong nonlinearity. We are aware that such an omission has an impact on the results; however, in this study, we intended to focus only on the viscoelastic behavior of the limb in flexion–extension. It is worth noting that for these two subjects, the negative values associated with the force counteracting elbow flexion are greater (considering absolute values) for the PD subject (Figure 5a) with respect to the healthy one (Figure 5b).

Figure 5c,d show the distribution of the RMSE between the double pendulum model and the $\beta$ coefficient model for the HC (Figure 5c) and for PD (Figure 5d) groups, respectively, with the possibility of appreciating the good fit since most of the subjects' RMSEs are limited to low values (<0.2).

Figure 6 presents boxplots illustrating the $\beta$ values under various rigidity conditions. Here, we define as "positive" the torque that induces a counterclockwise rotation around the shoulder's mediolateral axis, i.e., a forward oscillation of the arm. Consequently, we observe that $\beta_0$, representing the constant force associated with muscle activity, is positive. On the other hand, $\beta_1$, $\beta_2$, and $\beta_3$ exhibit negative values as they contribute to the generation of restoring forces, which act in opposition to changes in joint angles.

Figure 6a shows a trend for the forcing constant $\beta_0$ that mostly decrease (but not as marked) inversely with the MDS-UPDRS 3.3 item score. Indeed, PD groups with higher rigidity scores show smaller $\beta_0$ values, with statistical significance when discerning PD2 from HC ($p < 0.05$). As already stated, the $\beta_0$ coefficient represents the forcing constant due to muscle's activity. The progressive reduction in this coefficient, which proceeds according to disease severity, could be explained by the progressive loss of upper limb synkinesis of gait that occurs in PD patients starting in the early stages of the disease [42]. We obtained statistical significance comparing HC with PD02, that is, patients with more severe rigidity. These results suggest that our model could be more reliable in advanced disease stages.

Figure 6b evidences how the flexion elastic constant values $\beta_1$ do not have statistical significance since they do not result in particular differences among different groups of participants.

Figure 6c shows a decrease in the extension elastic constant $\beta_2$ with increasing PD severity. It is worth mentioning the differences between the flexion and extension elastic constants, $\beta_1$ and $\beta_2$, respectively, probably due to asymmetric effect of disease on flexors and extensors muscles.

Figure 6d reports differences in the dumping factor $\beta_3$ (associated with viscous behavior of the upper limbs) among groups. The differences are not so pronounced, but a downward trend with PD severity can be observed. This result suggests a possible impact of viscous-elastic factors in the limitation of upper limb movements in PD patients. Given that the viscous-elastic coefficient not associated with muscle contraction, we can hypothesize that $\beta_3$ could include the restraints on movement induced by ligaments, muscle sclerosis, and articular deformations. This interpretation could possibly explain the statistical significance obtained in the comparison between patients with higher rigidity scores (PD1, PD2) and HC. In fact, osteo-articular deformities are common comorbidities associated with more advanced stages of PD (i.e., antero/latero-collis, stooped posture, Pisa's syndrome) [24]. The progressive increase in viscoelastic forces could result in more abnormal and deformed postures that can eventually be associated with the origin of osteo-articular comorbidities.

Regarding the clinical–behavioral correlations, the results reported in Table 3 show a mild correlation between $\beta_0$ and the UPDRS-III rigidity score. This correlation supports the hypothesis that rigidity is partly responsible for reduced upper limb swing during gait in PD. The limited strength of this correlation, however, would suggest that other factors, such as bradykinesia, may also contribute to reduced arm swing during gait in PD. Arm swing reduction during gait has been recently associated with higher risk of falls in PD patients [43]. Relying on this model, objectively identifying PD patients with reduced arm swing and, therefore, patients prone to higher risk of falls could represent a valid instrument for fall prevention and treatment. For instance, applying our algorithm to

closed-loop systems, possibly with sensory cues to improve arm swing during gait, could represent a promising clinical application of this model.

When considering the present study, a number of possible limitations should be taken into account. First, the cohort of participants is rather limited, thus requiring the replication of findings in wider samples of patients, including those with more severe rigidity (i.e., UPDRS-III 3.3 item $\geq$ 3). Second, we assessed patients when in the OFF state of therapy, but arm swing and rigidity in PD can be significantly influenced by dopaminergic therapy [19,24]. Accordingly, future studies should address the effects of L-Dopa on the proposed model to clarify the impact of the pharmacological therapy on its accuracy.

Another limitation comes from condensing all the muscles acting on the limb in three coefficients only. The results suggest that this is a good approximation, since we are able to fit real data with a reduced error, but the model can be improved by differentiating the contribution of each muscle. Moreover, this model limits the analysis to planar movements, neglecting the arm's motion in three-dimensional space during oscillation. Conversely, the increase in complexity needs a more detailed analysis, so that it can be convenient to determine the best accuracy–complexity balance for a viable solution.

## 5. Conclusions

By considering that the swings of the upper limbs play a key, but often underrated, role during gait, here, we report a method for evidencing differences in the rigidity of forearm swings during gait for a group of PD patients with respect to a group of heathy subjects, for a total of 37 volunteer participants, and its relationship with the related MDS-UPDRS score.

To this aim, we model the arm swing as a double pendulum, focusing on the forearm movements as a key aspect, as the item 3.3 of the MDS-UPDRS rating scale evidences.

In particular, we use the double pendulum equations to derive the total torque acting on the forearm from data gathered using wearable sensors. The obtained torque is then described by means of a reduced set of coefficients representing the viscoelastic behaviour of the forearm. Accordingly, we differentiate swing aspects of the upper limbs during gait, evidencing a different viscoelasticity of PD patients with respect to their healthy counterparts.

Our results indicate a trend suggesting that the worse the pathology, the worse the capability to support walking with forearm swinging.

Moreover, the coefficients estimated through the model exhibit a correlation with the MDS-UPDRS item 3.3 scores, related to upper limb's rigidity.

In summary, our analysis demonstrates a correlation between UPDRS scores with respect to rigidity aspects of forearm swings. Nevertheless, we are aware that the number of involved participants must be improved to gain statistical effectiveness, that the model can be improved to take into account other second-order aspects in movements, and that other types of measurements, such as electromyographic ones, can add significance to the completeness of the study.

The model we propose, once validated in a more complex fashion too, could be associated with real-time evaluation of rigidity in PD patients in ecological settings. As the adoption of sensors and wearable devices increases in everyday clinical practice, our model could provide an easy tool for motor fluctuation detection and, therefore, for possible improvement in therapeutic strategies. Objective methods to assess tremor, bradykinesia, or other motor signs such as gait disorder (i.e., freezing of gait) are already largely available to use also in domestic environments. A new approach to quantitatively assess rigidity would help improve telemedicine strategies by implementing the remote examination of cardinal motor signs in PD through quantitative approaches.

**Author Contributions:** Conceptualization, L.P., F.G., C.M.V. and G.S.; methodology, A.S. and G.S.; software, L.P.; validation, G.S.; formal analysis, L.P. and G.S.; investigation, L.P., A.S., F.F., A.Z., M.P., V.R. and G.S.; resources, G.S. and A.S.; data curation, L.P.; writing—original draft preparation, L.P. and G.S.; writing—review and editing, G.S.; visualization, L.P.; supervision, A.S., F.G. and G.S.; project administration, G.S.; funding acquisition, G.S. All authors have read and agreed to the published version of the manuscript.

**Funding:** This research received no external funding.

**Data Availability Statement:** Data are available upon reasonable request.

**Acknowledgments:** We want to thank Captiks Srl for providing the wearable sensors.

**Conflicts of Interest:** One of the authors, G.S., declares involvement in Captiks Srl as a founder, but also declares to have not received money, goods, commodities, or any other advantages from this involvement other than scientific interest.

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
