# Peer review of "A Viscoelastic Model to Evidence Reduced Upper-Limb-Swing Capabilities during Gait for Parkinson’s Disease-Affected Subjects"

_electronics, doi:10.3390/electronics12153347_

Round 1

Reviewer 1 Report

General comment:

This manuscript describes a technology-based assessment of the forearm swing capabilities of Parkinson’s disease patients against healthy counterparts by means of a viscoelastic model. The authors tackle an inverse dynamic problem to determine the torque acting on the forearms. The work is relevant to the field of bioelectronic devices and systems with application to Parkinson’s disease, a worldwide chronic neurodegenerative disorder. Furthermore, the proposal is well-motivated and represents an advance in the knowledge for researchers and professionals working in bioelectronics. The experimental framework is clear and the results are well supported. The manuscript is interesting and well-written. Some points should be addressed before the manuscript can be accepted.

Comment 1:

In page 3, line 102, it should be “pc that receives” instead of “receive”. Also, check if pc (personal computer) was previously defined.

Comment 2:

In page 5, line 148, remove the comma before the word “and”.

Comment 3:

In Section 2.6:

-     -  The authors should give more information about the “smoothing algorithm” instead of only placing a reference.

-     -  The authors talk about a trust-region-reflective algorithm but do not mention what the algorithm is solving. Is it a least-squares problem? Please, clarify this situation.

-     -  The authors claim to the Spearman Rank correlation. Why this algorithm was selected? What do you are seeking with SRCA?

Comment 4:

The results in Fig. 5 c are nice. However, it lacks some discussion on the goodness of the fit. That is, how much does the estimated signal approximate the measurements?

Comment 5:

In the conclusions, the authors should include some future work besides the second-order statistics. For instance, there are plans to embed the algorithm in a wearable device? Could be useful a comparison with some machine learning algorithms?

Comment 6:

Carefully revise the manuscript for typos and grammatical errors.

Carefully revise the manuscript for typos and grammatical errors.

Author Response

Reviewer #1

Quality of English Language:

Yes

Can be improved

Must be improved

Not applicable

Does the introduction provide sufficient background and include all relevant references?

X

Are all the cited references relevant to the research?

X

Is the research design appropriate?

X

Are the methods adequately described?

X

Are the results clearly presented?

X

Are the conclusions supported by the results?

X

General comment:

This manuscript describes a technology-based assessment of the forearm swing capabilities of Parkinson’s disease patients against healthy counterparts by means of a viscoelastic model. The authors tackle an inverse dynamic problem to determine the torque acting on the forearms. The work is relevant to the field of bioelectronic devices and systems with application to Parkinson’s disease, a worldwide chronic neurodegenerative disorder. Furthermore, the proposal is well-motivated and represents an advance in the knowledge for researchers and professionals working in bioelectronics. The experimental framework is clear and the results are well supported. The manuscript is interesting and well-written. Some points should be addressed before the manuscript can be accepted.

Q1: Comment 1: In page 3, line 102, it should be “pc that receives” instead of “receive”. Also, check if pc (personal computer) was previously defined.

A1:  Thank you for your comment. The issue is solved.

Q2:  Comment 2: In page 5, line 148, remove the comma before the word “and”.

A2:  Thank you for your comment. The comma is removed.

Q3:  Comment 3: In Section 2.6: The authors should give more information about the “smoothing algorithm” instead of only placing a reference.

A3:  Thank you for your comment. We added the requested detail (page 6, line 232-238)

Q4:  In Section 2.6: The authors talk about a trust-region-reflective algorithm but do not mention what the algorithm is solving. Is it a least-squares problem? Please, clarify this situation.

A4:  Thank you for your comment. We added information on the algorithm (page 7, line 239-243)

Q5:  In Section 2.6: The authors claim to the Spearman Rank correlation. Why this algorithm was selected? What do you are seeking with SRCA?

A5:  Thank you for your comment. We added some explanations on how we use the Spearman Rank correlation coefficient (page 7, line252-25).

Q6:  Comment 4: The results in Fig. 5 c are nice. However, it lacks some discussion on the goodness of the fit. That is, how much does the estimated signal approximate the measurements?

A6:  Thank you for your comment. We discussed with more details the results   (page 10, line 306-317)

Q7:  Comment 5: In the conclusions, the authors should include some future work besides the second-order statistics. For instance, there are plans to embed the algorithm in a wearable device? Could be useful a comparison with some machine learning algorithms?

A7:  Thank you for your comment. We improved the conclusion (page 11, line 404-412).

Q8:  Comment 6: Carefully revise the manuscript for typos and grammatical errors.

A8:  Thank you for your comment. The paper is fully revised.

Reviewer 2 Report

While the introduction of the article provides a comprehensive overview of Parkinson's disease and its symptoms, there are a few negative aspects that could be improved:

Lack of Citations: The introduction includes references to previous studies in some parts, but many statements lack specific citations to support the information presented. Adding citations for relevant studies would enhance the credibility and validity of the claims made in the introduction.

Incomplete Information on Previous Studies: The introduction briefly mentions some previous studies that utilized technology-based objective measurements for PD assessment. However, it lacks sufficient detail to understand the specific findings and contributions of these studies to the field. More in-depth explanations and summaries of these studies would improve the context for the proposed research.

Absence of Research Questions or Hypotheses: While the introduction outlines the objective of exploring differences in the acting torque on the elbows between PD patients and healthy subjects, it does not explicitly state the research questions or hypotheses guiding the investigation. Including clear research questions would help readers understand the specific goals of the study.

Limited Discussion on Forearm Swing Significance: The introduction mentions the importance of forearm swing capabilities during gait for PD patients but does not elaborate on its significance. Providing a more in-depth discussion of why assessing forearm swings is relevant and how it can impact the understanding of PD motor impairments would strengthen the introduction's argument.

Lack of Information on the Proposed Model: While the introduction mentions the adoption of a descriptive model of forearm swings during gait, it does not provide any details about the model's features or how it will be applied in the study. Including a brief explanation of the proposed model would offer readers a better understanding of the study's methodology.

No Mention of Limitations or Scope of the Study: The introduction does not address the potential limitations of the proposed research or discuss its scope. Identifying and acknowledging potential limitations would demonstrate a more nuanced understanding of the study's boundaries and potential challenges.

Methods

Small Sample Size: The study has a relatively small sample size of 37 participants, with only 15 PD patients and 22 healthy subjects. A larger sample size could improve the statistical power and generalizability of the findings.

Inclusion and Exclusion Criteria: The inclusion and exclusion criteria for PD patients and healthy subjects are briefly mentioned, but they lack clarity and specificity. A more detailed description of these criteria is necessary to ensure that the study groups are well-defined and comparable.

Disease Severity Stratification: The study divides PD patients into subgroups (PD0, PD1, and PD2) based on the MDS-UPDRS scale, but the specific criteria for this stratification are not clearly explained. Providing a rationale for this division and specifying the criteria used to categorize patients would improve transparency.

Off-State Examination: The methods mention that PD patients were examined in the off state of therapy (after at least 12 hours of drug withdrawal). However, the rationale for choosing this specific timing is not provided. The potential impact of medication withdrawal on forearm swing capabilities during gait should be addressed and discussed

Assessment Tools: While the study mentions the use of MDS-UPDRS, H&Y, and MMSE scales for assessing PD patients, it does not well describe the specific motor and cognitive assessments conducted during the study. Providing details of the assessments and their relevance to the research objectives would enhance the methodology's comprehensiveness.

Model Simplification: The study adopts a simple model based on the 2-link arm inverse kinematics to represent the forearm's dynamic behavior. However, the rationale for choosing this specific model and the potential limitations of its simplifications should be discussed in the methods section.

Data Processing: The methods mention the use of a smoothing algorithm to increase the signal-to-noise ratio of the angles gathered during the TUG test. However, the specifics of this algorithm and its potential impact on the results should be detailed further.

Statistical Analyses: The methods state that a Mann-Whitney U test and Spearman Rank correlation analysis were conducted but do not provide specific details about the variables analyzed and their relationship to the research questions. A more detailed explanation of the statistical analyses performed would strengthen the methodology.

Potential Bias: The use of proprietary software for data recording and analysis, as well as the involvement of the software manufacturer (Captiks Srl) in the study, could introduce a potential conflict of interest or bias. The methods section should address any potential biases and assure the reader of the study's impartiality.

Discussion

Lack of Interpretation: The discussion provides a description of the findings from Figures 4, 5, and 6, but it lacks an in-depth interpretation of these results. The authors should provide more insights into the implications of the observed trends and how they relate to the study's objectives. A clear discussion of the clinical significance and potential applications of the findings is essential to enhance the impact of the study.

Insufficient Comparison with Existing Literature: The discussion does not adequately compare the study's results with previous research on forearm swings and rigidity in PD patients. A comprehensive literature review would help place the findings in context and highlight any novel contributions or deviations from existing knowledge.

Limited Generalizability: The discussion does not address potential limitations in generalizing the study's results to a broader PD population. The relatively small sample size, single-center recruitment, and specific patient characteristics may restrict the study's applicability to a wider range of PD patients.

Model Limitations and Assumptions: The discussion briefly mentions that certain constraints in the model were empirically considered of second order of importance. However, a more thorough discussion of the model's limitations, assumptions, and potential sources of error is necessary for readers to understand the study's reliability and accuracy.

Statistical Analysis Limitations: The discussion refers to significant correlations between MDS-UPDRS item 3.3 and certain β coefficients but does not elaborate on the strength of these correlations or their practical implications. 

Biological Interpretation Missing: The discussion lacks a biological interpretation of the β coefficients' changes with PD severity. Explaining how the changes in forcing constants and muscle viscosity may be related to the underlying pathophysiology of PD would add depth and relevance to the findings.

Discussion of Clinical Implications: The discussion does not sufficiently address the clinical implications of the study's results. How can the identified differences in β coefficients be used in practical clinical settings? Are they potential biomarkers for PD severity or treatment response? Such insights would enhance the discussion's practical relevance.

no

Author Response

Reviewer #2

Quality of English Language: (x) Moderate editing of English language required

Yes

Can be improved

Must be improved

Not applicable

Does the introduction provide sufficient background and include all relevant references?

X

Are all the cited references relevant to the research?

X

Is the research design appropriate?

X

Are the methods adequately described?

X

Are the results clearly presented?

X

Are the conclusions supported by the results?

X

While the introduction of the article provides a comprehensive overview of Parkinson's disease and its symptoms, there are a few negative aspects that could be improved:

Q1: Lack of Citations: The introduction includes references to previous studies in some parts, but many statements lack specific citations to support the information presented. Adding citations for relevant studies would enhance the credibility and validity of the claims made in the introduction.

A1:  Thank you for your comment. The introduction is improved and new citations are added.

Q2: Incomplete Information on Previous Studies: The introduction briefly mentions some previous studies that utilized technology-based objective measurements for PD assessment. However, it lacks sufficient detail to understand the specific findings and contributions of these studies to the field. More in-depth explanations and summaries of these studies would improve the context for the proposed research.

A2:  Thank you for your comment. We added more information about previous works.

Q3: Absence of Research Questions or Hypotheses: While the introduction outlines the objective of exploring differences in the acting torque on the elbows between PD patients and healthy subjects, it does not explicitly state the research questions or hypotheses guiding the investigation. Including clear research questions would help readers understand the specific goals of the study.

A3:  Thank you for your comment. The research questions/hypotheses are now stated in the “Introduction”

Q4: Limited Discussion on Forearm Swing Significance: The introduction mentions the importance of forearm swing capabilities during gait for PD patients but does not elaborate on its significance. Providing a more in-depth discussion of why assessing forearm swings is relevant and how it can impact the understanding of PD motor impairments would strengthen the introduction's argument.

A4:  Thank you for your comment, we improved the “Introduction” accordingly (page2, line 72-81)

Q5: Lack of Information on the Proposed Model: While the introduction mentions the adoption of a descriptive model of forearm swings during gait, it does not provide any details about the model's features or how it will be applied in the study. Including a brief explanation of the proposed model would offer readers a better understanding of the study's methodology.

A5:  Thank you for your comment, explanations are added accordingly (page 2, line 86-96, page 3, line 97-99).

Q6:        No Mention of Limitations or Scope of the Study: The introduction does not address the potential limitations of the proposed research or discuss its scope. Identifying and acknowledging potential limitations would demonstrate a more nuanced understanding of the study's boundaries and potential challenges.

A6:  Thank you for your comment. Statements concerning the study limitations are added (page 11, line 363-378).

Methods

Q7: Small Sample Size: The study has a relatively small sample size of 37 participants, with only 15 PD patients and 22 healthy subjects. A larger sample size could improve the statistical power and generalizability of the findings.

A7:  Thank you for your comment. As you evidenced with “Q6”, this work (as well as any other one in literature) has its limitations. We underlined this aspect in “Conclusions”. Anyway, because of the difficulty in enrolling PD patients available to participate in experimental procedures within a single institution, as it occurs for a non-negligible published works, from 10 to 20 PD patients is a frequent recurring number, as well as several referenced papers declare. Anyway, the overall number is now increased with respect to the previous version of the paper.

Q8: Inclusion and Exclusion Criteria: The inclusion and exclusion criteria for PD patients and healthy subjects are briefly mentioned, but they lack clarity and specificity. A more detailed description of these criteria is necessary to ensure that the study groups are well-defined and comparable.

A8:  Thank you for your comment. In line with your observation, we have now provided a more detailed description of inclusion and exclusion criteria (page 3, lines 107-118)

Q9: Disease Severity Stratification: The study divides PD patients into subgroups (PD0, PD1, and PD2) based on the MDS-UPDRS scale, but the specific criteria for this stratification are not clearly explained. Providing a rationale for this division and specifying the criteria used to categorize patients would improve transparency.

A9:  Thank you for your comment. We have now clarified the rationale and criteria used to classify PD patients into 3 subgroups (PD0, PD1, PD2) (page 3, lines 118-122)

Q10:      Off-State Examination: The methods mention that PD patients were examined in the off state of therapy (after at least 12 hours of drug withdrawal). However, the rationale for choosing this specific timing is not provided. The potential impact of medication withdrawal on forearm swing capabilities during gait should be addressed and discussed

A10:      Thank you for your comment. The impact of dopaminergic therapy on arm swing is discussed, both in the introduction and methods. Moreover, we also clarified that OFF-state of therapy is intended after at least 12 hours of drug withdrawal following standardized procedure which take into account Levodopa pharmacokinetics and duration of clinical effects. Finally we provided some references of previous studies adopting a similar approach to assess patients when OFF state of therapy  (page 3, lines 121-123)

Q11:      Assessment Tools: While the study mentions the use of MDS-UPDRS, H&Y, and MMSE scales for assessing PD patients, it does not well describe the specific motor and cognitive assessments conducted during the study. Providing details of the assessments and their relevance to the research objectives would enhance the methodology's comprehensiveness.

A11:      Thank you for your comment. We provided a comprehensive description of the adopted standardized clinical scales, internationally used in the clinical evaluation of PD patients. Moreover, we better specified their relevance to the research objectives (page 3, lines 125-133)

Q12:      Model Simplification: The study adopts a simple model based on the 2-link arm inverse kinematics to represent the forearm's dynamic behavior. However, the rationale for choosing this specific model and the potential limitations of its simplifications should be discussed in the methods section.

A12:      Thank you for your comment. We added some discussion on the limitation of the model (page5, line 197-201)

Q13:      Data Processing: The methods mention the use of a smoothing algorithm to increase the signal-to-noise ratio of the angles gathered during the TUG test. However, the specifics of this algorithm and its potential impact on the results should be detailed further.

A13:      Thank you for your comment. We added new details about smoothing algorithm (page 6, line 232-238)

Q14:      Statistical Analyses: The methods state that a Mann-Whitney U test and Spearman Rank correlation analysis were conducted but do not provide specific details about the variables analyzed and their relationship to the research questions. A more detailed explanation of the statistical analyses performed would strengthen the methodology.

A14:      Thank you for your comment.  We added new details about statistical analysis  (page 7, line 252-257)

Q15:      Potential Bias: The use of proprietary software for data recording and analysis, as well as the involvement of the software manufacturer (Captiks Srl) in the study, could introduce a potential conflict of interest or bias. The methods section should address any potential biases and assure the reader of the study's impartiality.

A15:      Thank you for your comment. No bias issues since the system was previously validated against a gold-standard one, as now underlined. Moreover, “Conflicts of Interest” is reported in the devoted Section (before the “References”).

Discussion

Q16:      Lack of Interpretation: The discussion provides a description of the findings from Figures 4, 5, and 6, but it lacks an in-depth interpretation of these results. The authors should provide more insights into the implications of the observed trends and how they relate to the study's objectives. A clear discussion of the clinical significance and potential applications of the findings is essential to enhance the impact of the study.

A16:      Thank you for your comment. We added some new statements on the clinical significance of the study findings (page 10, lines 326-331; 341-351)

Q17:      Insufficient Comparison with Existing Literature: The discussion does not adequately compare the study's results with previous research on forearm swings and rigidity in PD patients. A comprehensive literature review would help place the findings in context and highlight any novel contributions or deviations from existing knowledge.

A17:      Thank you for your comment. We improved the discussion of findings by also adding new references of previous works in the field.

Q18:      Limited Generalizability: The discussion does not address potential limitations in generalizing the study's results to a broader PD population. The relatively small sample size, single-center recruitment, and specific patient characteristics may restrict the study's applicability to a wider range of PD patients.

A18:      Thank you for your comment. As you evidenced with “Q6” and “Q7”, this work (as well as any other one in literature) has its limitations. We underlined this aspect in “Conclusions”. Anyway, because of the difficulty in enrolling PD patients available to participate in experimental procedures within a single institution, as it occurs for a non-negligible published works, from 10 to 20 PD patients is a frequent recurring number, as well as several referenced papers declare. Anyway, we slightly enlarged the number of subjects with respect to the previous version of the work.

Q19:      Model Limitations and Assumptions: The discussion briefly mentions that certain constraints in the model were empirically considered of second order of importance. However, a more thorough discussion of the model's limitations, assumptions, and potential sources of error is necessary for readers to understand the study's reliability and accuracy.

A19:      Thank you for your comment. We added a discussion about limitations (page10, line 306-310)

Q20:      Statistical Analysis Limitations: The discussion refers to significant correlations between MDS-UPDRS item 3.3 and certain β coefficients but does not elaborate on the strength of these correlations or their practical implications.

A20:      Thank you for your comment. We improved the discussion about the finding of mild correlation between MDS-UPDRS item 3.3 and β0 coefficient (page 10, lines 352-358)

Q21:      Biological Interpretation Missing: The discussion lacks a biological interpretation of the β coefficients' changes with PD severity. Explaining how the changes in forcing constants and muscle viscosity may be related to the underlying pathophysiology of PD would add depth and relevance to the findings.

A21:      Thank you for your comment. We improved the discussion concerning the biological interpretation of β coefficients (page 10, lines 326-331; 341-351)

Q22:      Discussion of Clinical Implications: The discussion does not sufficiently address the clinical implications of the study's results. How can the identified differences in β coefficients be used in practical clinical settings? Are they potential biomarkers for PD severity or treatment response? Such insights would enhance the discussion's practical relevance.

A22:      Thank you for your comment. We added more information, in the “discussion” and in the “conclusions” sections, concerning some clinical implication of the results of this study (page 11, lines 356-362 – page 11, lines 404-412)

Reviewer 3 Report

Issues related to the use of electronic components in the field of supporting treatment are topics of high priority.

However, the introduction should include information on how the authors of the paper propose to use the described solutions in the diagnosis and treatment of Parkinson's disease. The conclusions, on the other hand, lack directions for further development.

The article lacks information on existing solutions in the field of other wearable electronics systems

Under the equations, there is no description of the constants and variables used.

Author Response

Reviewer #3

Quality of English Language: (x) I am not qualified to assess the quality of English in this paper

Yes

Can be improved

Must be improved

Not applicable

Does the introduction provide sufficient background and include all relevant references?

X

Are all the cited references relevant to the research?

X

Is the research design appropriate?

X

Are the methods adequately described?

X

Are the results clearly presented?

X

Are the conclusions supported by the results?

X

Issues related to the use of electronic components in the field of supporting treatment are topics of high priority.

Q1: However, the introduction should include information on how the authors of the paper propose to use the described solutions in the diagnosis and treatment of Parkinson's disease. The conclusions, on the other hand, lack directions for further development.

A1:         Thank you for your comment. We improved comments about the contribution and application of our model to the diagnosis and treatment of Parkinson’s disease, and provided future developments of the presented model (please, also see the Q22)

Q2: The article lacks information on existing solutions in the field of other wearable electronics systems

A2:  Thank you for your comment. We added the information you asked in the “Wearable electronics” section.

Q3: Under the equations, there is no description of the constants and variables used.

A3:  Thank you for your comment, the issue is now solved accordingly.

Round 2

Reviewer 2 Report

Thank you for addressing the comments. No further modifications are needed.

All comments were addressed.

Reviewer 3 Report

After improving the paper is ready to be published